# Support of the Implementation of a Whistleblowing System for Smoke-Free Environments: A Mixed Methods Approach

**DOI:** 10.3390/ijerph182312401

**Published:** 2021-11-25

**Authors:** Al Asyary, Meita Veruswati, La Ode Hasnuddin S. Sagala, La Ode Ahmad Saktiansyah, Dewi Susanna, Hanns Moshammer

**Affiliations:** 1Department of Environmental Health, Faculty of Public Health, Universitas Indonesia, Depok 16424, Indonesia; alasyary@ui.ac.id (A.A.); dsusanna@ui.ac.id (D.S.); 2Study Program of Public Health, Faculty of Health Sciences, Universitas Muhammadiyah Prof. Dr. HAMKA, Jakarta 12130, Indonesia; meitaveruswati@uhamka.ac.id; 3Study Program of Information System, Faculty of Information and Technology, Universitas SembilanBelas November, Kolaka 93561, Indonesia; hasnuddin.sagala@gmail.com; 4Study Program of Public Health Science, Faculty of Public Health, Universitas Halu Oleo, Kendari 93232, Indonesia; saktiansyah89@gmail.com; 5Department of Environmental Health, Center for Public Health, Medical University of Vienna, 1090 Vienna, Austria; 6Department of Hygiene, Medical University of Karakalpakstan, Nukus 230100, Uzbekistan

**Keywords:** whistleblowing, smoke-free policy, law enforcement, Indonesia

## Abstract

Enforcement of a smoke-free policy is of vital concern in support of the health of smokers and bystanders. Indonesia has issued a smoke-free law, but implementation and enforcement lie with the regional and municipal governments. In a survey of 225 respondents recruited via schools, knowledge about the health effects of smoking and the smoke-free regulation, as well as attitudes towards and commitment and support of the enforcement of the smoke-free regulation in the Kendari City through an electronic whistleblowing system was examined. Furthermore, the participants were asked about the smoking status and smoking behavior. About half of the respondents were students (teenagers), the other half—their parents. Male respondents were strongly overrepresented (85%). Only 18% of the respondents declared to be smokers, mostly adults and males. Both the smokers and the non-smokers supported the smoke-free law and its enforcement through a whistleblowing system. Representatives of the local government were interviewed and participated in focus group discussions. In general, they also exhibited strong support of an electronic enforcement tool. However, issues of efficiency, costs, and responsibility must still be resolved. Nevertheless, an electronic whistleblowing system has the potential to further the health and livelihoods in a community like the Kendari City.

## 1. Introduction

Restricting smoking in public places sends a clear signal to smokers and helps them reconsider their vice before they fall victim to addiction. It protects non-smokers from passive smoking, especially in indoor and crowded settings [1]. It also helps denormalizing smoking behavior and has a strong protective effect on children and adolescents that otherwise are in great danger of smoking initiation [2].

The Indonesian regulation for smoke-free environments (SFEs) has the potential to reduce the harmful effects of smoking and especially to protect non-smokers from passive smoking in public places [3,4]. This regulation [5] focuses on health facilities, educational facilities, children playgrounds, places of worship, public transportation, workplaces, and public recreational places. It stipulates that active smoking in such public spaces is one of the acts against public health. However, this rule is considered to have a weak bargaining position because legal sanctions for violators are not specified in the national law but depend on detailed implementation through regional/regional or regency/city regulations. The principle of subsidiarity and the move to decentralization gave local leaders (regents and mayors) a special authority called regional autonomy to weigh and shape policies, including the Regional Regulation on SFEs in their regions [6,7].

The Kendari City aims to implement the concept of sustainable development by reducing negative impacts of societal and industrial development on health and the environment. Among the goals, one is to minimize or suppress the impact of cigarette smoking on the environment [8]. The Kendari City passed the SFEs regulation through the Kendari City Regional Regulation No. 16 dated 2014 [9]. However, until now, cross-related sectors have stated that there is no optimal indicator that can identify the reduction in the negative impact of cigarette smoking through regional regulations on SFEs.

In several other multisectoral problems, whistleblowing can be a viable solution based on evidence in disclosing violations or illegal acts, unethical/immoral, or other actions that can harm oneself or others [10]. The use of IT in whistleblowing allows disclosures to be made in secret (confidentially) against violations (smoking) [11]. It ensures timely information sharing and prompt reaction by the authorities in charge. It allows for more precise law enforcement in an accountable and transparent manner, for example, through photographic documentation. Various policy options can later be pursued either through guidance counseling for smokers in each unit (UKS, offices, etc.) or through the provision of civil sanctions. This study aimed to measure the support of students, their parents, and government officials of the development of whistleblowing systems (WBS) for smoking in the Kendari City.

## 2. Materials and Methods

### 2.1. Study Design

This research was conducted from November to December 2019 in the Kendari City, Indonesia. It was conducted with a mixed methods design, with a quantitative approach using a questionnaire directed at students and their parents and with a qualitative approach using in-depth interviews and focus group discussions (FGD) with local government subjects from the Health and Educational Office.

### 2.2. Quantitative Approach

The subjects were approached through schools (one junior and one senior high school) and consisted of students and their parents because Law No. 36 of 2009 on Health has a strong focus on schools. Sometimes, other household members besides the parents answered the questionnaire as well. The students were teenagers (age range: 11–19 years), the parents’ age ranged from 20 to 59 years.

To ensure sufficiently precise results [12], the minimum sample of this study was estimated to be around 200; however, to allow for possible non-responses, 240 subjects received a questionnaire measuring their direct intentions toward SFE implementation in their respective schools. The inclusion criteria were as follows: (a) students who were at school at the time of the study plus their parents, (b) domiciled in the City of Kendari, (c) willing to be subjects without coercion, and (d) owning (or using) a smartphone or a similar device. The latter criterion was necessary because the respondents were also asked about their willingness to use their phone for whistleblowing. In the second part of the study, they were also recruited into the pilot testing of the whistleblowing software. Informed consent was obtained from the respondents. For the students, the approval was also sought from the parents witnessed by the teacher.

For knowledge, commitment, and support as well as smoking behavior, we used the Guttman scale. Regarding knowledge, two questions were asked about the harmfulness of smoking (active and passive smoking), 11 questions—about the SFE regulation. The respondents had to indicate with an “X” sign whether a given statement was true or false. In the sections on support of and commitment to the SFE regulation, the respondents answered “yes” or “no”. The information on smoking behavior was also obtained through binary response options.

In the attitude section, using a four-point Likert scale, which is used to measure opinions and perceptions about social phenomena, the respondents selected “Strongly agree”, “Agree”, “Disagree”, or “Strongly disagree” with an “X” sign. An English translation of the questions is presented in Appendix A.

Descriptive results are presented as the absolute numbers and percentages or as the means, medians, and span, where appropriate. The smokers and the non-smokers were compared using t-tests. The associations between variables were first analyzed in univariate regression (linear regression for quasi-continuous outcomes and logistic regression for binary outcomes). The significance level was set to 0.05. The variables that exhibited significant associations were further examined in multiple regressions. Statistical analyses were performed using STATA BE Version 17.

### 2.3. Qualitative Approach

Interviews and FGDs were performed with five (IN1–IN5) high-ranking officers of the City Health Office, the City Education and Culture Office, and the City Satpol PP. At first, high-ranking officers were invited to a FGD. The official invitation letter ensured participation of representatives of all the relevant departments. The FGD began with a presentation about the current limitations of monitoring and evaluating every single violation of SFE and the impact of smoking on air pollution in the Kendari City. After that, the WBS-SFEs system was introduced and the stakeholder support and commitment was sought. After the FGD, in-depth interviews were conducted in private with each interviewee. The interviews and discussions were recorded and transcribed afterwards. The interview data were processed through content analysis. Statements were extracted from the transcript and grouped by two independent researchers that afterwards compared their grouping decisions. Exemplary statements were used to illustrate the opinions of each interviewee.

The study had been approved by the Ethics Committee of the Indonesian Association of Public Health Experts before the study began (No. 008/KEPK-IAKMI/XI/2019). All the participants, including the local government of the Kendari City, provided written consent for their data to be used for research purposes and the reporting of the study findings.

## 3. Results

### 3.1. Survey among Students and Parents

Among the students, only very few females fulfilled the inclusion criterion of a smartphone. In total, 225 of the 240 respondents returned a complete questionnaire (response rate of 93.8%). Of these, 119 (52.9%) were students (teenagers aged 11–19 years) and 106 (47.1%) were adults (mostly parents), 190 (84.4%) were male and 35 (15.6%) were female. Among the students, 117 were male and only two were female. Among the parents the genders were more equally distributed, but still fathers answered the questionnaire more often than mothers: 73 vs. 33 respondents.

Forty-one (18.2%) of the respondents declared to be smokers. Table 1 describes the characteristics of the respondents by smoking status.

The determinants of the smoking status are presented in Table 2. Adults and males were more likely to smoke than adolescents and women. These two associations remained significant in the multiple logistic regression analysis as well.

Those with the highest school grade “Junior high” reported the lowest smoking rates (12.4%) but only those with “Senior high” had a significantly higher smoking rate (25.4%). Furthermore, the few postgraduates reported a high smoking rate, but their number was too small for meaningful analysis. In adults, education is a predictor of both the attained knowledge and the socioeconomic status. In students, it also describes the knowledge attained, but is rather associated with the age than with the socioeconomic status. Therefore, the difference in smoking rates per educational status was additionally examined in the subgroup of adults only.

Among the professions, the first group (“not working”) consisted mostly of students. Their low smoking rate was likely not due to job characteristic, but due to the age category. As a consequence, the influence of the job category on smoking rates was only considered in the subgroup of adults. Entrepreneurs and general employees reported higher smoking rates than civil servants. However, the difference was only significant for entrepreneurs. Because of low numbers in the subgroups, educational level and profession were not further analyzed in multiple regression.

The respondents were asked about the health impact of smoking. Most confirmed that smoking is bad for health. However, at least 10 (4.5%) did not know or rejected the statement that both smoking and passive smoking pose a health risk. The smokers were more likely to disregard the dangers of smoking. Only five of the 184 non-smokers (2.7%), but also five of the 41 smokers (12.2%), negated the risk of passive smoking (*p* = 0.015). Although the students gave a wrong answer regarding health risks of smoking more often (seven times) than the adults (only three times), that difference was not significant. The ten persons giving a wrong answer were on average 21.7 years old, while the remaining 215 respondents aged 24.44 years on average. However, this difference was not significant.

Among the 11 questions about the SMF law, 1–11 were answered correctly (Figure 1). Although the single respondent who answered all 11 questions correctly was a student, on average, the adults answered more questions correctly (mean = 7.48) than the students (mean = 6.52). The average difference was 0.97 (*p* < 0.001). The smokers indeed gave on average more correct answers (7.3 of a possible total of 11) than the non-smokers (6.9). However, this difference was not significant.

The 41 smokers additionally answered questions about their smoking behavior (Table 3). Even the smokers agreed to a high extent (90.2%) to supporting and using a whistleblowing system. Nevertheless, 53.7% reported smoking indoors at home and partly even when family members or others are nearby. Negating the risks of passive smoking had practically no effect on inconsiderate smoking behavior.

Attitude (four questions, 0–12 points), commitment (two questions, 0–2 points), and support (four questions, 0–4 points) towards the SFE law were not predicted by the detailed knowledge of the law, but by the knowledge about the risk of passive smoking (Table 4). The smokers (mean: 7.3 correct answers) did not differ significantly (*p* = 0.184) from the non-smokers (6.9 correct answers) in their knowledge of the SFE law nor did they differ in their commitment (1.7 vs. 1.8 points) or support (3.0 vs. 3.2 points) for the whistleblowing system. They did express less (*p* = 0.003) attitude (8.27 vs. 9.29) though.

### 3.2. Interviews after the Focus Group Discussions

#### 3.2.1. Support from Each Stakeholder in the Development of WBS in SFEs

All the informants stated that they support the development of WBS for SFEs in the Kendari City. Reporting violations in the implementation of the SFEs regulation through an application is very much in line with the vision and mission of the city of Kendari, namely the creation of a city and home worthy of habitation. Technology supporting optimal outcomes is necessary for the implementation of any legislation, especially SFEs. Smoking in an area designated as an SFE is a criminal act and therefore should be enforced. Passive smokers are most disadvantaged and should receive protection from the state through legislation of SFEs. Examples expressing that support are quoted in the following:


*“Our task is to serve the mission and the vision of the City of Kendari. To that end, we support the introduction of this application that has already been successfully implemented in the Bogor area. We must enforce the law, but we have little means of control now.”*
—IN1


*“For in the Department of Education, we are already implementing the application in the said school. The “Healthy Schools” concept calls for a radius of 50 m free of cigarette smoke. The rules are clear, but we do not have means to control it. That’s why this application is very necessary.”*
—IN2


*“If smoking is observed in the area of SFEs, and it could harm others, sanctions should be given. This application enables us to do so, I am sure.”*
—IN3

#### 3.2.2. Commitment of Each Stakeholder in the Development of WBS in SFEs

The commitment and support from a variety of stakeholders is encouraging. Essentially, stakeholders expressed commitment and support fully as shown in the following excerpts:


*“For two years, we have had an internal rule in the Regional Health Office that prohibits smoking and violations are fined. If someone is observed smoking a photo is taken for documentation and the head of the office is informed. Thus, this (proposed) application system can be applied as its activity would support our goals. Still, its practical feasibility must be shown. Nevertheless, I am very committed to facilitating this application including the necessary budgetary support.”*
—IN1


*“Especially Satpol PP will act as the enforcement of SFE regulations. Hope we can decrease the prevalence of smokers in the City of Kendari, particularly at facility services of health, but in consequence also at home, where smoking now is still prevalent.”*
—IN2


*“The Satpol PP is already mandated with the enforcement of that law since April 2019. (…) Satpol surely is committed to enforce the law and that application will clearly support our personnel, as a study in Bogor City has clearly demonstrated.”*
—IN3


*“In schools, stewards traditionally control students. A student caught smoking is immediately sent to the guidance counseling room. But with teachers and visitors, the enforcement is complicated and we depend on the help of Satpol PP.”*
—IN5

#### 3.2.3. Barriers and Opportunities in the Development of WBS in SFEs

The factors that become obstacles in implementing WBS for SFEs in Kendari City are the limitations and availability of funds. A challenge regarding the application of this system is ensuring that it can reach all places that become SFEs and can be integrated as a whole, especially in the SFEs law enforcement agency, namely Satpol PP in Kendari City. Additionally, challenges remain in what actions will be carried out to offenders of SFEs legislation, as well as the ability of the implementation of these applications to decrease the prevalence of smokers in Kendari.


*“As usual, we have only limited funds that are already used to finance another planned agenda annually. However, there is a total 10% of limited funds for unexpected activities/programs from the special Kendari Government allocation funds. It can be prioritized as cost sharing with other potential funders e.g., private companies and universities.”*
—IN2

However, obstacles and challenges became opportunities for the implementation of the initial WBS for SFEs in Kendari. Implementation of the development of the WBS application for SFEs provoked a positive response from some of the agencies associated, including the Department of Health of Kendari, the Department of Education of Kendari, Satpol PP of the Kendari City, and the province of Southeast Sulawesi. The results of the FGDs, which were conducted with the Head of the Department of Health of Kendari, showed enthusiastic responses regarding the implementation of the initial WBS for SFEs. The Head of the Department of Health of Kendari expressed his desire to do a pilot project implementation of the WBS application for SFEs in the scope of the service of the Department of Health of Kendari in 2020. He claimed to be able to raise the necessary budget, at least partly through cost sharing and draft the agenda of activities together with other players in the implementation of the pilot project.

Great potential was also obtained from all cross-sectors in the application of WBS for SFEs in their work. With the latest regulations regarding remuneration systems based on performance, each cross-sector strongly wishes that weak reporting and enforcement of non-smoking areas can become agenda/performance activities that are calculated by remuneration for them. Kendari City Satpol PP is the leading sector in receiving reporting and in prosecution. The Kendari City Health Office can assist by acting as an extension with health promotion officers taking a persuasive approach to violators when there are reports of SFE violations.


*“What is the model for the development of the application of this? What immediately can be done in the prosecution of violators of SFE regulations and by anyone who would do the action? How large a scope of application is this? What can cover the entire SFE areas and is able to connect all governmental institutions in town?”*
—IN2.

## 4. Discussion

### 4.1. Characteristics of the Respondents

The group studied in the survey was not necessarily representative of the Kendari population, but represents an important target population because schools are the main focus of the SFE regulation. The majority of the subjects were teenagers (52.9%) and male (84.4%). The gender distribution likely reflects culturally founded roles and expectations. Of course, these cultural roles are subject to change, and they also differ between rural and urban communities. At present, women are still often less outspoken and engage less in public debates. Female children are less encouraged to participate in discussions at school or participation in surveys is less encouraged by their parents. On the other hand, these cultural roles still ensure that only few women smoke in Indonesia, as our data also demonstrated. It is to be feared that with the advent of emancipation, this beneficial side effect of female behavior perceived as “proper” will erode. These patterns in behavior change have been first observed in Western countries [13] and will likely also occur in Indonesia, if no countermeasures are introduced forcefully. Currently, adolescents receive special attention in efforts to reduce cigarette consumption [14,15]. Teens are very vulnerable to exposure and are at a crucial stage in the development of their character, knowledge, and attitude [16,17].

As the young generation will inherit the nation’s future, the smoking behavior of school teenagers today is very worrying [18]. The potential to become a captive market of industrial tobacco makes teenagers the focus of attention in the prevention and control of tobacco. The prevalence of smoking increased from 8% to 11% in the adolescent range of 10–18 years of age [19].

However, a teenager also has potential in computer literacy [20]. Teenagers are capable of quickly learning and utilizing every development of technology and information [21,22]. Adolescents’ time is frequently spent using technology that is also developing rapidly [23].

### 4.2. Smoking Behavior

About half of the active smokers reported smoking in the presence of others at work, at home, or in other places. Hence, there is significant potential for abuse of SFEs. Violation of SFEs occurs ranging from the smell of cigarette smoke in a designated region [24] to people visibly smoking. The Government of Kendari will need to work hard toward supporting social behavior which discourages smoking in SFEs.

Attention was specifically given to the implementation of SFEs in homes in Indonesia [25]. The study shows that the active smokers smoke at home (53.7%), which means that the home is not an SFE. Although it does not seem feasible and maybe not even desirable to enforce SFE in private homes, it is hoped that stricter regulation and enforcement of SFE in public places will also precipitate behavioral changes in the private settings.

### 4.3. Support and Commitment of Stakeholders in the Development of WBS in SFEs

The role of local governments in implementing SFEs is critical; this role must be demonstrated by commitment and real action. One of the tangible actions that can be done is the implementation of the WBS application as a form of law enforcement of the SFE legislation. The results of the study show that all the stakeholders are supportive and committed to developing WBS applications for SFEs. Development of the WBS application is considered to be an easy achievement to ensure Kendari is worthy of habitation.

To follow up on this research, the Health Office will coordinate with relevant stakeholders in the development of the Kendari City SFE WBS application. The stakeholders who will also participate in the development of the system are the Department of Communication and Information, the Department of Education, Satpol PP, and the Secretariat of the Regions.

### 4.4. Support of SFE Enforcement through the WBS Application

Enforcement of SFEs is essential to the Kendari City’s community. WBS enables enforcement of the SFE law, including effective monitoring and evaluation of internal groups abusing smoking rules, particularly at educational facilities. Additional work is needed to collaborate with local governments, including educational facilities, to implement this system as well as to look at the ability to measure its impact in decreasing active smoking in the youth. Furthermore, our data show that even when smoking in public places is banned, there are still many smokers that report smoking at home and/or in front of their family members. Overall, about half of all the smokers reported such behavior. Therefore, more efforts are needed to educate smokers and inform them about the health risks their addictive behavior poses on their family.

Smoke-free policies work [26] and are supported by non-smokers, but also, to an astonishingly high extent, by smokers themselves [27,28]. Enforcement is key and novel IT technology can support enforcement of the SFE legislation [29,30]. Implementation of such tools is still controversial and many practical issues like sharing of costs and responsibility, proper use, and adequate response to the surge of information still need to be resolved. This was also expressed by the officers during the interviews. Nevertheless, support of a WBS in Kendari was high both among non-smokers and smokers, and also among the high-ranking officers interviewed. This provides hope for future policy implementation in the field of smoking protection.

## 5. Conclusions

In the Kendari City, there is strong support from the public and from administration to implementing WBS to enforce SFE in public places, especially at schools. In the next step, such a system will be implemented. This needs careful planning and clear rules as to the responsibilities of the different administrative units and the sharing of the initial costs of implementation. This endeavor also needs strong political support from the local governments and ongoing scientific evaluation. However, in general, the results of our study allow for a positive outlook.

## Figures and Tables

**Figure 1 ijerph-18-12401-f001:**
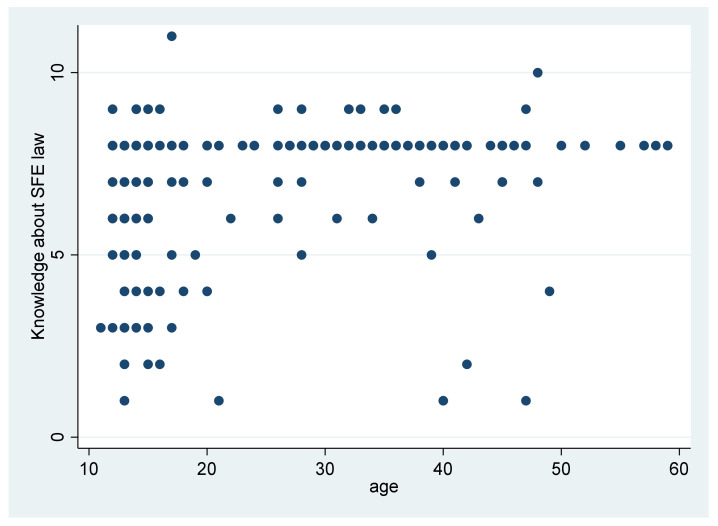
The adults on average answered more questions on the SFE law correctly.

**Table 1 ijerph-18-12401-t001:** Description of the participants of the survey.

Characteristics	Total (*n* = 225)	Smokers (*n* = 41)	Non-Smokers (*n* = 184)
Age group			
Students (11–19 years)	119 (52.9%)	14 (34.1%)	105 (57.1%)
Adults (20–59 years)	106 (47.1%)	27 (65.9%)	79 (42.9%)
Gender			
Male	190 (84.4%)	40 (97.6%)	150 (81.5%)
Female	35 (16.6%)	1 (2.4%)	34 (18.5%)
Highest education			
Elementary school	10 (4.4%)	2 (4.9%)	8 (4.3%)
Junior high school	89 (39.6%)	11 (26.8%)	78 (42.4%)
Senior high school	59 (26.2%)	15 (36.6%)	44 (23.9%)
Undergraduate	61 (27.1%)	11 (26.8%)	50 27.2%)
Postgraduate	6 (2.7%)	2 (4.9%)	4 (2.2%)
Profession			
Student/unemployed	147 (65.3%)	20 (48.8%)	127 (69.0%)
Civil servant	64 (28.4%)	13 (31.7%)	51 (27.7%)
Entrepreneur	9 (4.0%)	5 (12.2%)	4 (2.2%)
Worker	5 (2.2%)	3 (7.3%)	2 (1.1%)

**Table 2 ijerph-18-12401-t002:** Determinants of the active smoker status.

	Active Smoker		Bivariate Logistic		Multiple Logistic	
	Yes	No	*p*	OR (95% CI)	*p*	OR (95% CI)
Age						
Student	14 (11.8%)	105 (88.2%)		Ref.		Ref
Adult	27 (25.5%)	79 (75.5%)	0.009	2.56 (1.26–5.21)	<0.001	4.08 (1.96–8.51)
Gender						
Female	1 (2.9%)	34 (97.1%)		Ref.		Ref
Male	40 (20.0%)	150 (78.9%)	0.032	9.07 (1.20–68.27)	0.006	17.59 (2.28–135.85)
Education					N/A	N/A
Elementary	2 (20%)	8 (80%)	0.502	1.77 (0.33–9.45)		
Junior high	11 (12.4%)	78 (87.6%)		Ref.		
Senior high	15 (25.4%)	44 (74.6%)	0.045	2.42 (1.02–5.72)		
Undergrad.	11 (18.0%)	50 (82.0%)	0.337	1.56 (0.63–3.87)		
Postgrad.	2 (33.3%)	4 (66.7%)	0.171	3.55 (0.58–21.69)		
Education (adults only)					N/A	N/A
Elementary	1 (14.3%)	6 (85.7%)	0.243	0.26 (0.33–2.47)		
Junior high	1 (100%)	0 (0%)	Dropped (only one case)			
Senior high	12 (38.7%)	19 (61.3%)		Ref.		
Undergrad.	11 (18.0%)	50 (82.0%)	0.034	0.35 (0.13–0.92)		
Postgrad.	2 (33.3%)	4 (66.7%)	0.804	0.79 (0.31–5.01)		
Profession (adults only)					N/A	N/A
No work	7 (23.3%)	23 (76.7%)	0.739	1.19 (0.42–3.39)		
Civil servant	13 (20.3%)	51 (79.7%)		Ref.		
Entrepreneur	4 (57.1%)	3 (42.9%)	0.045	5.23 (1.04–26.33)		
Employee	3 (60.0%)	2 (40.0%)	0.066	5.88 (0.89–38.95)		

**Table 3 ijerph-18-12401-t003:** Smoking behavior among the 41 smokers (14 students, 27 adults).

Smoking Behavior			Passive Smoking Is Harmless	
	Yes (%)	No (%)	*p*	OR (95% CI)
Smoking indoors at work	18 (43.9)	23 (56.1)	0.851	0.833 (0.124–5.606)
Smoking indoors at home	22 (53.7)	19 (46.3)	0.235	4.0 (0.406–39.367)
Smoking near family	19 (46.3)	22 (53.7)	0.762	0.745 (0.111–5.007)
Smoking near others	18 (43.9)	23 (56.1)	0.116	6.286 (0.636–62.162)

**Table 4 ijerph-18-12401-t004:** Attitudes, commitment, and support.

	Passive Smoking Is Harmless			
	Yes (10)	No (215)	*p*	Difference
Attitude (0–11)	6.7	9.22	0.0001	2.52
Commitment (0–2)	1.3	1.83	0.0003	0.53
Support (0–2)	2	3.25	0.0002	1.25

## Data Availability

All the datasets generated and/or analyzed during this study are not publicly available due to confidentiality, but are available on reasonable request.

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
