# Peer review of "Support of the Implementation of a Whistleblowing System for Smoke-Free Environments: A Mixed Methods Approach"

_ijerph, 2021, doi:10.3390/ijerph182312401_

Round 1

Reviewer 1 Report

Overall, I found this manuscript to present an innovative solution to the challenge of enforcing smoke-free policies to have a meaningful impact on reducing tobacco use and protecting all for the harmful effects of cigarette smoke. I have some recommendations for the authors to consider in improving the presentation of their methods, results, and conclusions.

Line 1 – May consider using the term mixed-methods in title

Line 39 – I recommend reconsidering the wording of the first sentence of the introduction. Rather than considering smoking simply a vice, the role of tobacco use addiction needs to be considered.

Line 47 – It would be interesting to know more details about the specifics of the regulation. Is it smoking in any indoor and outdoor public space? Are there any designated areas or distance specifications within the national policy?

Line 48-49 Check grammatical and spelling errors here.

Line 71 – This statement of study purpose is quite broad. I would encourage a statement of a student and parent population. Your study does not broadly collect from the Kendari City population. Are you focused specifically on reducing youth smoking and overall enforcement of the smoke-free policy in educational facilities?

Line 76 – The description of the study design with quantitative and qualitative parts may be considered a mixed methods approach. I would not consider the study design observational. It is not directly observing any smoking behaviors or policy compliance.

Line 78 –I would recommend there be a parallel approach to the description here. If you include the population for the qualitative approach, also include the population for the quantitative approach.

2.2 Quantitative Approach – This section may be clearly if it is separated into student respondents and parent respondents. The age group for students should be clearly identified.

Lines 93-96 – The presentation of informed consent is confusing to read. I recommend separating the process for students and parents.

2.3 Qualitative Approach

Line 117-118 – Who presented information to the officers? How did the research team eliminate response bias? The officers may have felt pressured to give answers that were socially acceptable based on the presentation. Would officers of lower rank feel pressured to answer to officers of higher rank?

Line 125 – Exemplary statements illustrate the opinion of the specific respondent, and cannot represent the opinions of all interviewers unless all officers agreed and expressed a similar statement.

  1. Results Section

Line 132-144 I would include the language used to describe the results align with characteristics in the Table. In the table, mother and father terminology is not used. Be consistent throughout this section. It may be useful to breakdown demographics for students and parents separately. It is confusing to know the educational levels of the student respondents in Table 1 and Table 2.

Line 145 – For Table 1, would not consider Students as an age group – more specific ages for these groups would be valuable.

Line 158-166 Reporting these results by students and parents would be more helpful to the reader.

Line 174 Is this table including just the adult/parent respondents?

Lines 185-253 3.2 Interviews and focus groups  

What, if any, different information did you collect in the focus group versus the interviews? Does the IN1-5 represent an interview #? Did you have any illustrative quotes from the focus group? Were there any differences in the officers, by rank or region?

Line 254-296  4. Discussion  I would like to see a richer, more in-depth discussion. The whistle-blowing system is an innovative idea, but how can it be implemented to have a meaningful impact that doesn’t cause harm. What about the issues of efficiency, cost, and responsibility that were mentioned in the study abstract?  This section could be more aligned to the study purpose and results. The information on adolescents and technology needs to be tied to use of the whistle-blowing system. I would recommend organizing into discussion regarding student respondents, parent respondents, and government officials.  It also needs to include more tie-ins with other recent literature about smoke-free environment policy compliance. More current smoke-free environment enforcement-compliance references would be beneficial. A section noting the limitations would be useful. Five seems a small number for the focus groups and interviews. How do you know that you reached saturation of opinion? Also, what are potential unintended consequences of the whistle-blowing system?  

Line 258. The use of teenagers is not used previously in the results section. Would clarify the age group of students and align to m A subject is missing with the 84%.

Line 259. The gender distribution is interesting. More elaboration on the cultural roles, expectations, and gender perceptions of smoking would be valuable.

Line 297-304 Conclusions

This conclusion paragraph does not reflect overall findings reflected in the study abstract.

Author Response

please find our responses in the attached file ("reviewer1")!

Reviewer 2 Report

Introduction

  1. Line 63-65 “On the other hand, the utilization of information technology (IT), another aim of the city of Kendari, is still not optimal, especially regarding the individual reporting function (whistleblowing)”. Why is this sentence mentioned here? Does this sentence fit the main purpose of the introduction?

  1. Line 68-69, The use of IT in whistleblowing allows disclosure to be made in secret (confidential) against violations (smoking). Is this the only justification for using IT in whistleblowing? In other words, could authors provide more advantageous points to use IT in whistleblowing?

Materials and Methods

  1. In Line 104, “In the attitude section using the Likert scale, which is used to measure opinion…”. Please specify that it is 4-point liker scale here

  1. In Line 109, “Two groups were compared by t-tests”…. What two groups were compared here, please specify it.

  1. Also for the author should state what software were used for both qualitative and quantitative analysis.

Results

  1. Page 7, In subsection “Barriers and opportunities in the development of the WBS in SFEs”. In line 223, the author talked about “The factors that become obstacles in implementing the WBS for SFEs in Kendari City are the limitations and availability of funds”. Could author provide some quotes to spell out this challenge?

Discussions

  1. In page 7, line 258, the author mentioned “the majority of the subjects were teenagers (52.9%) and (84.4%)”. why there are two percentages for teenagers. Could author explain this?

  1. In page 7, from line 268-271, the author mentioned “However, a teenager also has potential in computer literacy…”. Why this character of teenager is important for this study? Could author further explain it?

Author Response

Please find our responses in the attached file!
